# CRISPR/Cas9 Editing of Duck Enteritis Virus Genome for the Construction of a Recombinant Vaccine Vector Expressing *ompH* Gene of *Pasteurella multocida* in Two Novel Insertion Sites

**DOI:** 10.3390/vaccines10050686

**Published:** 2022-04-27

**Authors:** Nisachon Apinda, Yongxiu Yao, Yaoyao Zhang, Vishwanatha R. A. P. Reddy, Pengxiang Chang, Venugopal Nair, Nattawooti Sthitmatee

**Affiliations:** 1Department of Veterinary Biosciences and Veterinary Public Health, Faculty of Veterinary Medicine, Chiang Mai University, Chiang Mai 50200, Thailand; nisachon_a@cmu.ac.th; 2The Pirbright Institute, Ash Road, Woking GU24 0NF, UK; yongxiu.yao@pirbright.ac.uk (Y.Y.); yaoyao.zhang@pirbright.ac.uk (Y.Z.); vishi.avalakuppapapireddy@pirbright.ac.uk (V.R.A.P.R.); pengxiang.chang@pirbright.ac.uk (P.C.); venugopal.nair@pirbright.ac.uk (V.N.); 3Jenner Institute, University of Oxford, Oxford OX1 2JD, UK; 4Department of Zoology, University of Oxford, Oxford OX1 2JD, UK; 5Excellence Center in Veterinary Bioscience, Chiang Mai University, Chiang Mai 50200, Thailand

**Keywords:** Cre-Lox, CRISPR-Cas9, duck enteritis virus, fowl cholera, NHEJ, *Pasteurella multocida*, viral vector

## Abstract

Duck enteritis virus (DEV) and *Pasteurella multocida*, the causative agent of duck plague and fowl cholera, are acute contagious diseases and leading causes of morbidity and mortality in duck. The NHEJ-CRISPR/Cas9-mediated gene editing strategy, accompanied with the Cre–Lox system, have been employed in the present study to show that two new sites at UL55-LORF11 and UL44-44.5 loci in the genome of the attenuated Jansen strain of DEV can be used for the stable expression of the outer membrane protein H (*ompH*) gene of *P. multocida* that could be used as a bivalent vaccine candidate with the potential of protecting ducks simultaneously against major viral and bacterial pathogens. The two recombinant viruses, DEV-OmpH-V5-UL55-LORF11 and DEV-OmpH-V5-UL44-44.5, with the insertion of ompH-V5 gene at the UL55-LORF11 and UL44-44.5 loci respectively, showed similar growth kinetics and plaque size, compared to the wildtype virus, confirming that the insertion of the foreign gene into these did not have any detrimental effects on DEV. This is the first time the CRISPR/Cas9 system has been applied to insert a highly immunogenic gene from bacteria into the DEV genome rapidly and efficiently. This approach offers an efficient way to introduce other antigens into the DEV genome for multivalent vector.

## 1. Introduction

There are numerous diseases that affect and cause substantial mortality to duck populations all over the world. Fowl cholera, caused by the bacterium *Pasteurella multocida*, and duck virus enteritis (DVE), also known as duck plague caused by duck enteritis virus (DEV), have been recognized and studied for many years and remain important diseases affecting ducks [1]. Available vaccines showed variable results in protecting ducks against fowl cholera in natural outbreaks [2,3]. The outer membrane protein H (OmpH), a major membrane protein of the *P. multocida* envelop [4] and a cross protective antigen among avian *P. multocida* strains [5,6,7], incorporated into flow cholera vaccine candidate, provided the strong protective immunity against *P. multocida* in chickens [5,6,8,9], ducks [10,11], and cattle [12,13]. Meanwhile, the live attenuated DEV vaccines are widely used for reduction of diseases impact in ducks [3]. The whole genomes of DEV are approximately 158 kbp in length and contain 78 predicted open reading frames (ORFs) of putative proteins [14]. Due to its large genome and limited host range, DEV has been exploited as a vector for development of recombinant multivalent vaccines [15,16,17,18]. In Thailand, the attenuated DEV Jansen strain, from embryonated chicken eggs, has been routinely used as live vaccines in ducks for over half a century. In a comparison of the genome sequences DEV Jensen and 2085 strain with partial sequences, NCBI BLAST showed 100% identity among two open reading frames (ORFs) in the LORF11 (GenBank accession no. JQ430740.1) and LORF3 (GenBank accession no. JQ43078.1) across both strains. The whole genome of the 2085 strain (GenBank accession no. JF999965.1) was represented the genome of the Jensen vaccine strain in this study.

Virus vector vaccines have advantages over traditional vaccines through the induction of both cellular and humoral responses, enabling differentiation between infected and vaccinated hosts [19]. Viral vectored vaccines, particularly using vectors, such as adenovirus, herpesvirus, and poxviruses, are used widely in veterinary medicine [20,21,22,23,24]. Several approaches have been applied for the construction of recombinant DEV vaccine candidates against poultry viruses. These include the traditional methods of homologous recombination, or vector engineering approaches using bacterial artificial chromosome (BAC) and fosmid systems [25]. However, many of these methods for generating recombinant DEV are often less efficient, time-consuming, and labor-intensive, especially due to the requirement of several rounds of plaque purification and transfer vector cloning procedures [19]. Recent advances in genome editing using clustered regularly interspaced short palindromic repeats associated protein nuclease 9 (CRISPR/Cas9) have helped in the rapid insertion of foreign antigens into the genomes of several viruses [19,25,26,27,28,29,30]. Compared with traditional methods, CRISPR/Cas9 editing also provides an opportunity to produce multivalent recombinant vaccines for simultaneous protection against major avian diseases [19,25,27,28,29,31]. The programmable nuclease Cas9, directed by a single-guide RNA (sgRNA), can introduce double-strand breaks (DSBs) in target sites of genomic DNA. Then, there are two key methods used for gene insertion: NHEJ (non-homologous end joining) and HDR (homology-directed repair pathway) [32]. Although HDR is attractive because of its high fidelity [33], mammalian cells preferentially employ NHEJ over HDR as NHEJ is active throughout the cell cycle, whereas HDR is restricted to S/G2 phases. NHEJ is also faster than HDR, and it is also known to suppress the HDR process [34]. The combination of CRISPR/Cas9 and Cre–Lox recombination techniques provides a powerful and versatile system to excise pre-determined LoxP sites, allowing the excision of specific genetic fragments and selectable markers [26,35]. Through its simplicity and effectivity, CRISPR/Cas9 technology has shown to be beneficial for gene modification and provides an alternative to recombinant vaccine construction.

The strategy the duck industry needs is to develop a combination of live attenuated vaccines to induce simultaneous protection against multiple diseases through the reduced administration of a multivalent vectored vaccine at reduced cost to improve poultry welfare. Previously, the effectiveness of mixed immunization against *P. multocida* and DEV was demonstrated [10]. However, new recombinant vaccines with the potential of inducing stronger humoral and cellular immune responses against these major pathogens are important. This study was aimed to identify new insertion site(s) in DEV genome (Jansen strain) for the stable expression of foreign genes using the advanced NHEJ-CRISPR/Cas9 and Cre-Lox system. We also evaluated the effect of insertion sites on the stable antigen expression and growth ability of the recombinant virus that are needed for the further development of a recombinant DEV vaccine that can simultaneously express *ompH* gene of *P. multocida* for simultaneous protection against virus-bacterial co-infection. The study also evaluated the choice of insertion site on antigen expression and growth ability of the vector virus.

## 2. Materials and Methods

### 2.1. Cell Culture and Virus

Primary chick embryo fibroblasts (CEF), from 10-day old embryos, were prepared and maintained in M199 medium (Thermo Fisher Scientific, Waltham, MA, USA) supplemented with 5% fetal bovine serum (FBS, Sigma, St. Louis, MO, USA), 100 units/mL of penicillin and streptomycin (Thermo Fisher Scientific), 0.25 mg/mL Fungizone (Sigma), and 10% tryptose phosphate broth (Sigma).

DEV Jansen strain, obtained from the Bureau of Veterinary Biologics Department of Livestock Development, Ministry of Agriculture and Cooperative Thailand, was used for the construction of the DEV recombinant candidate and DEV wild type.

### 2.2. Construction of sgRNAs Plasmid

The individual single guide RNA (sgRNA) targeting each region of the DEV genome, totalling 9 sgRNA targeting sites, was designed using CRISPR guide RNA designing software (http://crispor.tefor.net/org, accessed on 9 December 2019). Each sgRNA target sequence has ruled out any possible off-target sequences and cloned into the CRISPR/Cas9 vector pX459-v2 (Addgene plasmid #62988, Cambridge, MA, USA) by introducing synthesized oligo-DNA primers corresponding to the target sequence into BbsI cloning sites. The sg-A sequence was taken from published data [28] and cloned into px459-v2 in the same way. All sgRNA targeting sequences in DEV genome shown in Table 1.

### 2.3. Construction of the Donor Plasmid Containing the GFP-OmpH-V5 Expression Cassette

To generate a donor plasmid containing the ompH-V5 expression cassette tagged with a removable GFP reporter cassette shown in the Figure 1, the oligo pairs GFP-SgA-F and GFP-SgA-R (containing sg-A target sequence at both ends, a PacI site flanked with two LoxP sequences for GFP reporter cassette cloning and excision in the middle, and two SfiI sites for the cloning of the ompH-V5 expression cassette) were annealed and cloned into the pGEM-T-easy vector. Then, the GFP expression cassette was released by PacI restriction digestion from pEF-GFP and cloned into the resulting vector via the PacI cloning site, generating donor plasmid called pGEM-SgA-LoxP-GFP. Meanwhile, for the construction of ompH-V5 expression cassettes, we first cloned two SfiI sites into pcDNA3.1(+) via NdeI and PmeI by annealing the oligo pairs SfiIx2-F and SfiIx2-R generating pcDNA3.1(+)-SfiI. The *ompH* sequence (Accession No: U50907.1) tagged with V5 was synthesized into pTwist vector (Twist Bioscience, San Francisco, California, USA). Then, the ompH-V5 cassette was released from pTwist-OmpH-V5 and cloned into pcDNA3.1(+)-SfiI by NotI restriction sites generating pcDNA3.1(+)-OmpH-V5. Finally, the ompH-V5 expression cassette was transferred into pGEM-sgA-LoxP-GFP via SfiI, generating a complete donor plasmid called pGEM-sgA-LoxP-GFP-OmpH-V5. The primer sequences used for guide RNA cloning and donor plasmid construction are listed in Table 2. The plasmid DNA preparation of the donor plasmid, SgA and Cas9/gRNA expression plasmid DNAs, were extracted using a commercial DNA extraction kit (Qiagen) according to the manufacturer’s instructions.

### 2.4. Generation of Recombinant DEV-GFP-OmpH-V5 by NHEJ-CRISPR/Cas9-Mediated Gene Insertion

Primary CEF cells were plated into 12-well plates the day before transfection. Hence, 0.25 mg of each sgRNA targeting site and SgA were co-transfected with 0.5 mg donor plasmid into CEF cells using the TransIT-X2 Dynamic Delivery System (Mirus Bio, Madison, WI, USA) in accordance with the manufacturer’s instructions. At 24 h post transfection, the CEF cells were infected with DEV at a multiplicity of infection (MOI) of 0.1 plaque forming units (pfu)/cell. The infected CEF were harvested 48 h later and continued to the plaque purification by fluorescence-activated cell sorting (FACS).

For virus plaque purification, CEF cells were washed once with phosphate-buffered saline (PBS) before the infection with the DEV-OmpH-V5 from stock virus. The inoculum was removed at 2 h post-infection and replenished with either fresh medium or 2% Minimum Essential Medium (MEM)–agarose overlay.

### 2.5. Fluorescence-Activated Cell Sorting for Plaque Purification of the Recombinant Virus

Prepare two 96-well plates preseeded with 2 x 10^4^ CEF cells per well the day before sorting. Then 48 h post-infection, the transfected/infected CEFs were trypsinized, resuspend and transfer into a 1.5 mL microcentrifuge tube with 50 μL of FBS. Centrifuge at 200× *g* for 5 min. Resuspend the cells in 1 mL of PBS with 1% FBS. Count the cell numbers using a hemocytometer and adjust the number of cells to 1 × 10^6^ cells/mL. After that, transfer the cells to a polystyrene sorting tube through its strainer cap. Sort the single cells expressing GFP into 96-well plates seeded with CEFs using the cell sorter according to the manufacturer’s instruction. Incubate the sorted cells for 3 d at 38.5 °C with 5% CO_2_.

### 2.6. The Excision of the GFP Cassette from DEV-GFP-OmpH-V5 with Cre Treatment

For the excision of GFP expression cassette using Cre recombinase, 1 mg of pcDNA3-Cre was transfected into CEF in 12-well plates pre-seeded on the day before. At 24 h post transfection, the cells were infected with MOI 0.01 of DEV-GFP-OmpH-V5 at 24 h post transfection [31]. Two days later, the supernatant was kept and used to infect fresh CEF cells in 6-well plates and then overlayed with 2% agar-MEM to get the plaque without GFP. The excision of GFP from the virus by Cre recombinase treatment was confirmed by 3′ junction PCR using primer pairs OmpH-3F and LORF11-R, which amplifies the junction between OmpH cassette and LORF11, and other primer pairs OmpH-3F and UL44-F, which amplifies the junction of ompH cassette and UL44.. The primer sequences used for PCR described above are listed in Table 3.

### 2.7. Western Blot Analysis

The expression of OmpH protein in recombinant virus-infected CEF cells was determined by Western blot analysis using mouse polyclonal anti-OmpH as modified from the method as described previously [36,37]. Briefly, 2.5 × 10^6^ primary CEF were seeded into T25 flask the day before infection. The following day, the parental virus and each recombinant virus were diluted with growth medium to MOI 0.01 and added to the flask were then incubated at 38.5 °C and 5% CO_2_. At 48 h post infection, more cytopathic effects (CPE) were shown. The infected CEF cells were collected and boiled with TruPAGE LDS sample buffer (Sigma) for 7 min. The samples were separated on a 4–12% TruPAGETM Precast Gel, and the resolved proteins were transferred onto PVDF membranes. Immunoblots were blocked with 5% skimmed milk, then incubated with anti-OmpH primary antibodies (1:5000 dilution). After probing with primary antibodies, the blots were incubated with secondary antibody IRDye680RD goat anti-mouse IgG (LI-COR) and visualized using Odyssey Clx (LI-COR). On the other hand, duck serum positive DEV was used as primary antibody for DEV loading control using the same strategy.

### 2.8. Indirect Immunofluorescence Analysis (IFA)

The expression of V5-tag in recombinant virus-infected CEF cells was evaluated by immunofluorescence assays using immunocytochemistry. CEF cells grown in 24-well plates were infected with parental virus and each recombinant virus at MOI 0.01 for 48 h before harvesting. After fixing with ice cold acetone:methanol (1/1) for 10 min, the V5-tag expression was analysed using monoclonal mouse anti-V5 monoclonal antibody (Bio-Rad) in dilution 1:1000 followed by rabbit anti-mouse IgG labelled with Alexa Fluor 488 (Invitrogen) in dilution 1:200 to detect V5-tag expression. Moreover, the DEV-infected cells were detected with anti-DEV polyclonal rabbit serum in dilution 1:200 followed by goat anti-rabbit IgG labelled with Alexa Fluor 568 (Invitrogen, Waltham, MA, USA) in dilution 1:200, respectively. The positive stained cell images were taken using an IncuCyte in 36 separate regions per well per sample.

### 2.9. Stability of the Inserted Genes in the Recombinant Viruses

The recombinant viruses DEV-OmpH-V5 were grown sequentially in CEF cells for 15 passages. The expression of V5-tag was examined after every 5 passages by IFA and the integrity of the *ompH* gene insert was examined using DNA extracted from every 5 passages by PCR with outside primer pairs located at the flanking region of the insertion site as previously described. The primer sequences used for PCR described above are listed in Table 3.

### 2.10. Growth Properties of the Recombinant DEV-OmpH-V5

To determine the growth kinetics of the recombinant DEV-OmpH-V5 compared with DEV wild type, CEF cells were infected at an MOI of 0.01 of each virus. Then, the supernatants were harvested at different time points after infection (12 h, 24 h, 48 h, 72 h), and viral titers were determined by plaque assay.

### 2.11. Statistical Analysis

Statistical analysis was performed using GraphPad Prism 6 (GraphPad Software, La Jolla, CA, USA). Paired student *t*-test and one-way ANOVA were used to test differences between different groups. *p* values < 0.05 were considered significant.

## 3. Results

### 3.1. Targeted Knock-in of GFP-OmpH-V5 Expression Cassette into to Multiple Sites of DEV Genome Using CRISPR/Cas9 System

We first aimed to determine whether CRISPR-Cas9 system can be used to knock-in a foreign gene in nine intergenic regions located in the forward and reverse DNA strand of DEV genome without any potential adverse effects on viral replication. For this, we attempted to knock-in the GFP-OmpH-V5 expression cassette into nine distinct intergenic regions of DEV genome using different sgRNA targeting plasmids (Table 1) to examine the suitability of each insertion site to tolerate the foreign gene expression cassette in the recombinant DEV vector. After 48 h infection with DEV (0.1 MOI), the number of positive green plaques or areas of infected cells were analysed using fluorescent microscope as well as IncuCyte S3 Live-Cell Image analyser. In the determination based on the highest numbers of GFP-positive green infected cell areas, the sgRNAs targeting the UL55-LORF11, followed by UL44-44.5 intergenic regions of DEV genome, appeared to be the most efficient (Figure 2).

Moreover, the GFP-positive DEV plaques generated using the UL55-LORF11 and UL44-44.5 targeting sgRNAs continued to express GFP after repeated plaque purification compared to the other sgRNAs where the GFP expression disappeared during plaque purification. Based on these results, UL55-LORF11 and UL44-44.5 intergenic regions were selected for the CRISPR/Cas9-based gene editing for the generation recombinant DEV-OmpH-V5 in this study (Figure 3A). The two individual sgRNAs were designated as gRNA-7 and gRNA-9, respectively.

### 3.2. Generation of Recombinant DEV-GFP-OmpH-V5 Using NHEJ-CRISPR/Cas9-Mediated Gene Insertion

The donor plasmid that carries the GFP reporter gene cassette flanked by the SgA target sites [28] was used for the generation of GFP-OmpH-V5 expression fragment for designed integration. Two plasmids of the SgA for targeting the donor plasmid to release the insert fragment and gRNA-7 and gRNA-9 for targeting viral genome at UL55-LORF11 and UL44-44.5 intergenic regions, respectively, were cloned into pX459-v2 that expresses the codon-optimized S. pyogenes Cas9 (Cas9) as a bicistronic mRNA with the puromycin-N-acetyltransferase gene and the vectors expressing sgRNAs driven by the U6 promoter. The concurrent cleavage processing of donor plasmid DNA and the viral genome targeted by Cas9 result in the insertion of the GFP-OmpH-V5 cassette into the UL55-LORF11 and UL44-44.5 loci, as shown in Figure 3B.

Twenty-four hours post co-transfection of the donor plasmid with SgA and each Cas9/sgRNA vector, CEF cells were infected with DEV-wild type (DEV-WT) at MOI 0.01. Observation of GFP positive plaque 48 h post infection indicated successful insertion of OmpH-V5 expression cassette into the DEV genome. After that, the supernatant and cells were harvested and used to infect fresh CEF cells for plaque purification. Furthermore, the single cell fluorescence-activated cell sorting (FACS) technique was also applied to facilitate virus purification. Through CRISPR/Cas9 mediated recombination, the single cells positive for GFP signals were sorted into a 96 well plate pre-seeded with CEF cells. The purity of progeny GFP-positive virus, detected 48 h post-sorting, was confirmed by PCR using primers specific for the DEV insertion sites (UL55-F & LORF11-R and UL44-F & UL44.5-R). As shown in Figure 4b, the band specific for the DEV wild type was not detected by PCR in the sorted population of cells were infected with purified GFP virus. While the larger of the PCR fragments of GFP-OmpH-V5 cassette did not show in this time causes related to cycling conditions, this may not be sufficient for longer target sequences. However, this inserted cassette will be confirmed with 5′ and 3′ junctions in the next step. This purified recombinant virus is termed DEV-GFP-OmpH-V5, as indicated in the Figure 3C.

The insertion of GFP-OmpH-V5 cassette at each locus was double confirmed by PCR with 5′ and 3′ junction site-specific primers (Figure 4a) and sequencing of the PCR products. Positive PCR bands on 3′ junction site were observed in all the samples except DEV-WT and the H_2_O negative control (Figure 4c). Moreover, PCR with primers OmpH-F and OmpH-V5-R confirmed the presence of the full OmpH-V5 insert in each recombinant DEV-GFP-OmpH-V5 infected CEF (Figure 4d). Details of the primers used are as shown in Table 3.

Sequencing results revealed that insertion occurred in both sense (UL55-LORF11) and anti-sense (UL44-44.5) directions. Some of the clones of both insertion sites did show some indels, but this did not affect the open reading frame of the inserted OmpH-V5 gene expression cassette (data not shown). Taken together, these results demonstrated the usefulness of the CRISPR/Cas9 system as a powerful tool for the rapid generation of recombinant DEV.

### 3.3. Excision of the GFP Cassette from DEV-GFP-OmpH-V5 Using Cre Recombinase

Excision of the GFP expression cassette from DEV-GFP-OmpH-V5 virus was achieved using Cre recombinase expressed from pcDNA3-Cre construct transfected into CEF 24 h after infection (MOI 0.01) with DEV-GFP-OmpH-V5 virus. GFP-negative virus plaques could be observed 48 h post infection, with the removal of GFP cassette from more than 70% of the DEV-GFP-OmpH-V5 virus genome compared to those cells not having Cre plasmid transfection-infection (Figure 5). Infected tissue culture supernatant virus was harvested and used to infect fresh CEF cells for further selection and purification of GFP-negative plaques using overlay with 2% agar-MEM. GFP-negative virus plaques were further picked up and purified to obtain recombinant virus stocks. The purified recombinant virus stocks after the excision of the GFP cassettes were further confirmed to have the OmpH-V5 cassette by primers at both ends of each size and junction PCR with specific primers of each insertion site, as described previously in Figure 4a (data not shown). As expected, the GFP expression cassette was deleted from the virus stocks while the OmpH-V5 cassette was retained. The resulting recombinant viruses were redesignated as DEV-OmpH-V5 of UL55-LORF11 and DEV-OmpH-V5 of UL44-44.5, as indicated in the Figure 3D.

### 3.4. Characterization of the Recombinant DEV-OmpH-V5 Expressing OmpH-V5 Cassettes

As shown in Figure 6a, lower, the expression of OmpH protein was examined by Western blotting with anti-OmpH polyclonal mouse antibody using lysates of cells infected with DEV-WT or each recombinant DEV-OmpH-V5. The recombinant OmpH protein was used as positive loading control. As expected, the cell lysates from both recombinant DEV-OmpH-V5 infected cells (loaded in duplicate lanes) demonstrated 39.5 kDa OmpH-V5 protein identical to the positive OmpH protein control, which was absent in the cell lysates infected with DEV-WT. As shown in Figure 6a, upper, the antibody specific to DEV as a DEV loading control showed a positive band of all viruses.

The expression of V5-tag in each recombinant virus-infected CEF cells was evaluated by immunofluorescence assays (IFA). CEF cells infected with DEV-OmpH-V5 of UL55-LORF11 and UL44-44.5 showed clear V5-specific positive staining (green) and DEV-specific polyclonal serum positive (red) staining. As expected, CEF cells infected with parental DEV showed positive red staining for the polyclonal serum, but not with the V5 antibody (Figure 6b).

The replication property of each recombinant DEV-OmpH-V5 was determined by virus plaque size and multistep replication kinetics compared to the DEV-WT. CEF cells were infected with DEV-WT and DEV-OmpH-V5 of UL55-LORF11 and DEV-OmpH-V5 of UL44-44.5 (MOI 0.01) 48 h post-infection were used for the determination of plaque sizes. Measurement of the average diameters of at least 10 plaques of each virus did not show significant differences in the plaque sizes among all three viruses (Figure 6c). Similarly, a comparison of the multi-step replication kinetics of the three viruses did not show significant differences between both recombinants (DEV-OmpH-V5 of UL55-LORF11 and UL44-44.5) and the parental viruses (DEV-WT), with the titers reaching approximately 10^7^ PFU/mL at 72 h post-infection (Figure 6d). Thus, the insertion of the OmpH-V5 expression cassette in either of the loci did not appear to compromise the replication ability of the recombinant DEV-OmpH-V5 compared to the DEV wild type.

### 3.5. Stability of the Inserted Genes in the Recombinant Viruses

To determine the stability of the ompH-V5 expression cassette during serial passage, each recombinant DEV-OmpH-V5 was passaged continuously 15 times on CEF cells and viral DNA was extracted from the supernatant and analyzed after every five passages using outside primers of the insertion site (Figure 4a) that amplified the full length insert. As is clear from the PCR results (Figure 7a), the recombinant virus infected cells consistently showed the predicted full-length product at each of the passage levels. The absence of a lower band in the recombinant virus, accompanied with no variability in the size of ompH-V5 gene in during serial passaging of both recombinant viruses in cultured cells, indicated that the ompH-V5 gene was stably inserted into the DEV genome both insertion sites.

We have also examined the V5 expression every five passages of 15 serial passages by IFA (Figure 7b). Once the insert was lost from any of the recombinant viruses during serial passages, the cell staining would only show positive labelling with the anti-DEV antibody. Here, the detection of doubly stained cells demonstrating the expression of both antigens in the recombinant DEV confirmed the stable integration of the foreign gene. There was also no apparent effect on the replication ability of the recombinant virus during all sequential passage. These results indicated that ompH-V5 gene was stably integrated in both insertion sites of the DEV genome without any adverse effects on viral replication.

## 4. Discussion

Previous studies considering an *ompH* gene of *P. multocida* identified it as highly protective gene and suggested it to be a candidate vaccine for fowl cholera in duck which induced an efficient antibody response, able to reduce the degree of adhesion of the *P. multocida* strain to the duck embryo fibroblast cells, along with successfully inducing high levels of lymphocyte proliferation, providing clinical protection [10,11].

The DEV genome has been exploited as a recombinant viral vaccine vector using conventional reverse genetics techniques on full length BAC or Fosmid clones of the DEV genome [15,17,32,33]. While these tools are valuable for the development of recombinant vaccines, BAC recombineering and Fosmid library approaches face many challenges, including the significant time needed for the insertion of foreign gene inserts and subsequent removal of the BAC plasmid [27]. Moreover, the viral genome in the bacteria is often unstable [27], causing difficulty to generate difficultly recombinant viruses. Compared to this, the CRISPR-Cas9-based gene editing method described in this study is much faster and very efficient saving significant amount of time because the DNA double-strand cleavage to insert a foreign gene and rejoin cleaved DNA cleavage will be completed on the virus genome directly. Recently, the number of studies on animal viral vector vaccine development utilizing the CRISPR/Cas9 system with insertion or deletion of the gene has been increasing because of its high efficiency, specificity, versatility, flexibility, simplicity, and low cost compared to the other viral genome editing techniques [34]. A majority of the recent studies utilized successfully CRISPR/Cas9 for recombinant vaccines focusing on foreign viral gene expression on several viral vectors, such as herpesvirus of turkey (HVT) [27,28,29], infectious laryngotracheitis virus (ILTV) [30], and duck enteritis virus (DEV) [19,31]. This present study is the first report that demonstrates the effective use of the CRISPR/Cas9 and Cre-Lox system for the development vaccine opportunities with the potential for simultaneous protection against two major viral and bacterial pathogens that cause acute contagious diseases in ducks.

Previously, the two gene junctions UL27/UL26 and US7/US8 of DEV strain C-KCE were shown to be suitable for the double insertion of the foreign gene for the construction of recombinant C-KCE construction based on the CRISPR/Cas9-mediated gene editing strategy with the HDR pathway [19]. Whereas the junctions UL27/UL26 of DEV genome strain C-KCE can also be used for single insertion generating recombinant DEV via NHEJ with Cre-Lox system, the stability of foreign gene has not been examined [31]. However, the insertion of foreign genes into non-essential genes may partly affect properties of the parental virus and expression of foreign antigens [38]. Therefore, it is important to identify other sites in DEV genome where foreign genes can be inserted and expressed stably without disrupting properties of the DEV parental virus, which provides new appropriate sites for the future development of a multivalent DEV vaccine. Interestingly, the present study explored several genomic loci in the DEV genome for new stable site by CRISPR/Cas9 system. At least, our study has successfully identified the two new insertion sites of DEV genome for the stable expression of foreign genes using NHEJ-CRISPR/Cas9 gene editing. The PCR and immunostaining confirmed that the foreign gene (an *ompH* gene of *P. multocida*) could be integrated into two new targeting stable sites of DEV Jensen strain intergenic region (UL55-LORF11 and UL44-44.5) and maintained as far as 15 passages, indicating viral genetic stability. Furthermore, the fact that the recombinant viruses with foreign gene insertions had similar growth kinetic and plaque size compared to the wild type virus. These results are consistent with the results from previous studies. In 2014, Weng et al. constructed the recombinant DEV-H5 using BAC at UL55-LORF11 of DEV strain C-KCE. This HA expression cassette affects neither the growth kinetics of the virus nor its protection against DEV [32,33]. Whereas another previous study that successfully constructed recombinant virus v2085-H5ΔgC using BAC clone in lieu of the UL44 locus, the resulting foreign gene insertion effected viral titer significant reductions greater than 700-fold and plaque sizes increased in vitro when compared to the parental virus [17]. In contrast, the recombinant DEV with the foreign gene inserted in the UL55-LORF11 and UL44/44.5 by CRISPR/Cas9 did not alter pattern of variability in titers and plaque size when compared to parental of DEV Jansen strain. These findings further confirm the suitability both of UL55-LORF11 and UL44/44.5 sites as a foreign gene insertion for the construction of DEV-based vaccines. The off-target effects are still a major concern in the main limitations of the CRISPR technologies and can result in base deletion or insertion (indel) after repair, in turn resulting in a frame shift mutation [39]. However, developing a well-optimized and engineered CRISPR system can significantly reduce the off-target effects. For instance, off-target effects can be reduced via increasing the nucleases cleavage specificity or reducing the duration of Cas9 activity. Continuous efforts to understand all their pitfalls, improving editing capabilities, and making advances in the delivery systems will ensure the CRISPR system for the full potential to benefit society in near future [40].

As described, we have generated DEV recombinants with a foreign gene insertion of UL55-LORF11 and UL44-44.5 in either orientation by junction PCR, using two primers located at each end of the inserted sequence to identify in sense orientation and side swapping to identify the insert in antisense orientation following previous suggestion about the limitation of these applications [29]. Moreover, the beneficial effect of using the donor plasmid with an excisable GFP marker flanked between LoxP site has been re-substantiated in this study using different insertion sites of different strain in the same virus (DEV) or between different virus vectors (HVT and DEV) [28,31].

These experiments are now in progress to evaluate the possibility of multiple gene loci in the DEV genome as a foreign gene stable insertion site for the future development of DEV vectored vaccine using the powerful gene editing technique CRISPR/Cas9 and Cre-Lox system. This study also suggests approaches for the rapid construction of multiple recombinant vaccines with the potential for simultaneous protection against viral and bacterial diseases of ducks. The same application could be a very promising candidate locus for the rapid development of a series of bivalent or trivalent viral vector vaccines, which is expected to be of great benefit in controlling and protecting against virus and bacterial co-infection in duck diseases.

## 5. Conclusions

In the present study, DEV genomes of Jansen strain were edited to harbor and express specific bacterial antigens (an *ompH* gene of *P. multocida*) in the UL55-LORF11 and UL44/44.5 using NHEJ/CRISPR-Cas9 system. Apparently, *ompH-V5* gene expression from the knock-in cassette was not impacted by the orientation of insertion. Furthermore, those two insertion sites have been well verified as stably expressing the *ompH* gene of *P. multocida* and suggested to further develop DEV vector recombinant vaccine. This may become an alternative viral–bacterial bivalent vaccine strategy in the future. However, further assessment of the efficacy and protection of this recombinant vaccine is warranted in animal experiment.

## Figures and Tables

**Figure 1 vaccines-10-00686-f001:**
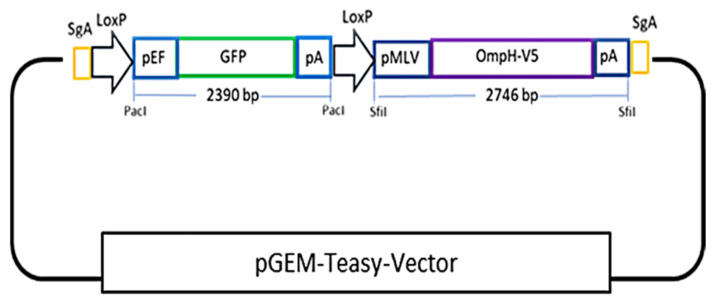
A schematic representation of the cloning strategy for donor plasmid construction containing the OmpH-V5 expression cassette tagged with a removable GFP reporter cassette.

**Figure 2 vaccines-10-00686-f002:**
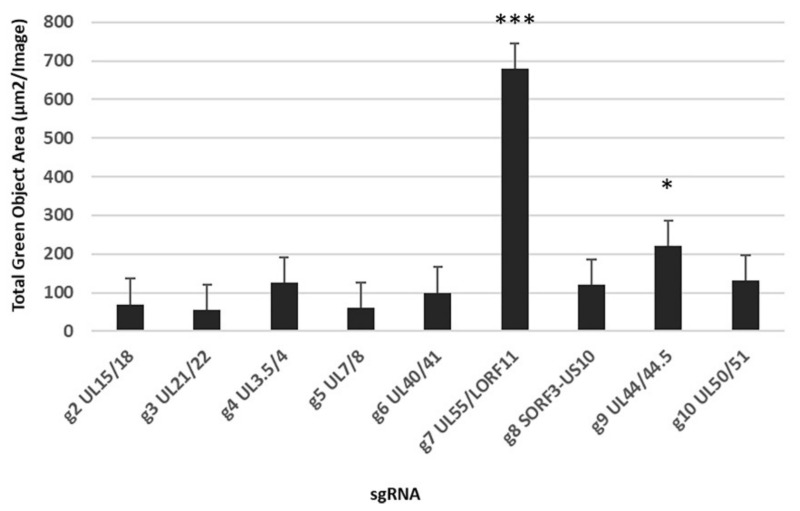
The total positive green cell area measurement with multiple gRNA target sites in first plaque purification of DEV-OmpH-V5. Error bar = standard error of mean. (* *p* < 0.05, *** *p* < 0.001).

**Figure 3 vaccines-10-00686-f003:**
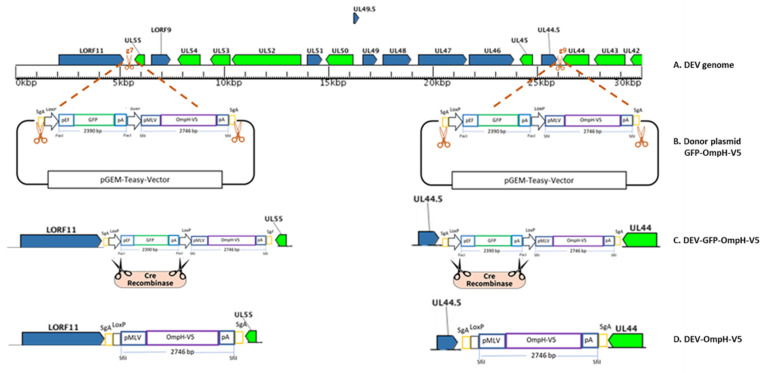
Schematic illustration of the recombinant DEV-OmpH-V5. (**A**) Full-length of the attenuated commercial DEV vaccine strain used in this study. The scissor icon represents the gRNA targeting sites in the intergenic region of DEV genome (UL55-LORF11 and UL44-44.5) for OmpH-V5 gene insertion via CRISPR-Cas9 base gene knock-in. (**B**) Two portions of the genome selected to insert fragment of the GFP and the OmpH-V5 expression cassettes released by Cas9/sgA cleavage from donor plasmid. (**C**) The recombinant DEV expressing the reporter GFP and OmpH-V5 before the GFP gene is excised by Cre recombinase. (**D**) After GFP excision, the recombinant vaccine candidate terms are DEV-OmpH-V5 of UL 55-LORF11 and UL44-44.5.

**Figure 4 vaccines-10-00686-f004:**
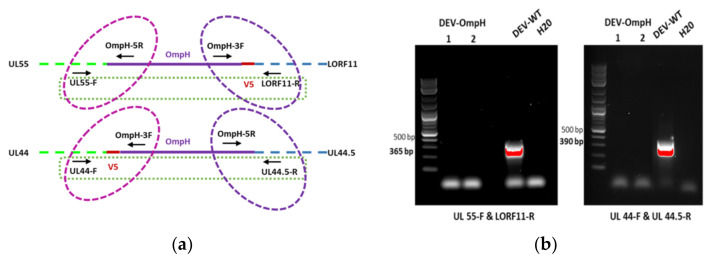
Verification of the recombinant DEV-OmpH-V5 by PCR. (**a**) The name and position of primers use for verification purification (Square shape) and 5′ and 3′ junction PCR (Oval shape). (**b**) The purification of DEV-GFP-OmpH-V5 recombinant virus in each insertion site and DEV wild type were detected by PCR. The primers pair between each intergenic region of DEV genome labelled under the panel. (**c**) 3′ junction PCR verification of DEV-GFP-OmpH-V5 with specific primers in each site to confirm integration and identification insertion sites. Sense orientation insertion of DEV-GFP-OmpH-V5 of UL55-LORF11 was detected by primer pair OmpH-3F & LORF11-R and anti-sense direction of UL44-44.5 was detected by primer pair OmpH-3F and UL44-R. (**d**) The confirmation of whole OmpH-V5 gene inserted in each recombinant DEV comparing with DEV wild type was identified by PCR using primers OmpH-F and OmpH-V5-R. The molecular size of DNA is indicated in the left lane.

**Figure 5 vaccines-10-00686-f005:**
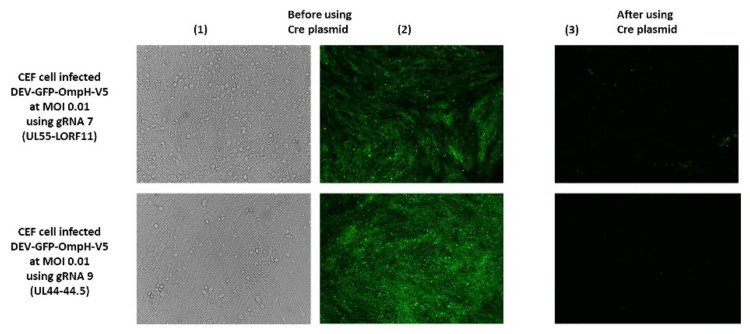
The Excision of the GFP Cassette from DEV-GFP-OmpH-V5 by Cre-Lox system. Before Cre plasmid addition, DEV-GFP-OmpH-V5 plaque under were observed under bright-field (**1**) and fluorescent microscopy (**2**). After Cre plasmid treatment (**3**), this panel shows more than 70% of GFP cassette was removed from the DEV-GFP-OmpH-V5 virus genome. fluorescent microscopy.

**Figure 6 vaccines-10-00686-f006:**
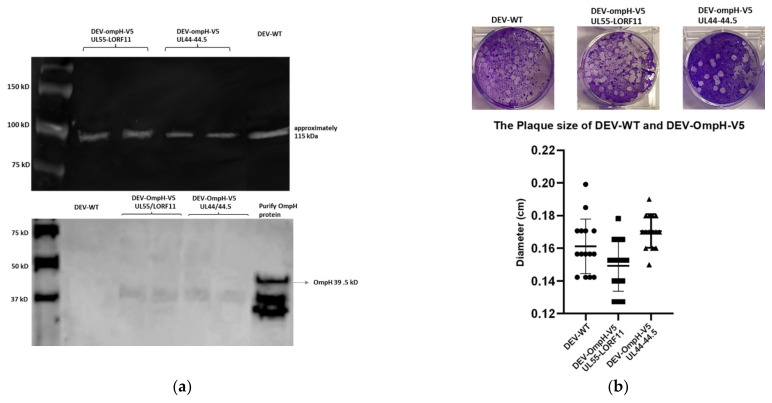
Characterization of each recombinant DEV-OmpH-V5 infected CEF cells. (**a**) Lower panel shows the expression of OmpH protein was analyzed by Western blot using anti-OmpH polyclonal antibody. The positive OmpH expression of CEF cell lysate infected with rDEV-Omp-V5 of UL55-LORF11 (lane 3 and 4) and rDEV-Omp-V5 of UL44-44.5 (lane 5 and 6) were duplicate loaded into each lean. While CEF cells lysate infected with DEV-WT was loaded into lane 2 for negative control. The recombinant OmpH protein was loaded in last lane for positive control. The molecular weight was indicated in first lane. Upper panel shows antibody specific to DEV as a DEV loading control. (**b**) This panel shows the confirmation the successful of V5 expression by indirect immunofluorescence assay (IFA) with anti-V5 monoclonal antibody (green) sequentially strained with DEV-infected mouse serum (red) for detection of DEV infected-cells. The region of merged images was taken by Incucyte machine. (**c**) Upper panels display plaque morphology of DEV wild type, DEV-OmpH-V5 of UL55-LORF11 and DEV-OmpH-V5 of 44-44.5. Lower graph shows the plaque diameters of CEF cells infected with recombinant DEVs and DEV wild type were measured at 6 days post infection. (**d**) The multi-step growth kinetic curve of recombinant DEVs and DEV wild type infected CEF cells at MOI 0.01. Supernatants were collected and viral titers were determined at the indicated time points post-inoculation by plaque assay. (*p* < 0.01).

**Figure 7 vaccines-10-00686-f007:**
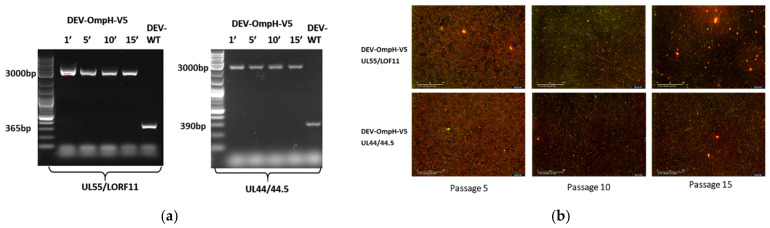
Stability of recombinant DEV-OmpH-V5. (**a**) Detection of OmpH-V5 gene insertion in DEV-OmpH-V5 was confirmed by PCR using primer at both ends of each insertion site during serial 5 passages compared with DEV-WT infected CEF as controls, indicating that the recombinant virus is stable. The lane numbers represent the passage numbers of the recombinant virus. (**b**) Detection of V5 expression of recombinant DEV was assessed by IFA. Merged images of positive double stained DEV-OmpH-V5 infected CEF with anti-V5 (Green) and anti-DEV (Red) was taken by Incucyte at 5th, 10th, and 15th passages.

**Table 1 vaccines-10-00686-t001:** The sgRNA lists and target sequences in DEV * for generation recombinant virus.

sgRNA ID	Target Sequences	PAM	Gene Locus
sgA	GAGATCGAGTGCCGCATCAC	CGG	SgA
g2	CAAGACAGACAAGTATTGCT	TGG	Between UL 15 & UL 18
g3	TGCGTAATTTCATAACCAAA	AGG	Between UL 21 & UL 22
g4	ATTTACGTCCTCGGGGGAGG	GGG	Between UL 3.5 & UL 4
g5	TTATTTCAAATATTAGTGTG	AGG	Between UL 7 & UL 8
g6	TTGGTAATCAAGAGTTTACT	GGG	Between UL 40 & UL 41
g7	GACTATGTAAAGACAGTCGA	CGG	Between UL 55 & LORF 11
g8	GTTTGCAATCCTTTATACAT	TGG	Between SORF3 & US 10
g9	GCACAACTTCAAAAATGATG	GGG	Between UL 44 & UL 44.5
g10	ACAACCTCTTCATATTAGAT	AGG	Between UL 50 & UL 51

* Genbank accession number: JF999965.1.

**Table 2 vaccines-10-00686-t002:** Primer sequences used for sgRNA plasmid and donor plasmid construction.

Primer	Sequences
GFP-SgA-F	GAGATCGAGTGCCGCATCACCGGATAACTTCGTATAATGTATGCTATACGAAGTTATTTAATTAAATAACTTCGTATAATGTATGCTATACGAAGTTATGGCCGCCTAGGCCGGCGCGCCGTTTAAACGGCCATTATGGCCGAGATCGAGTGCCGCATCACCGG
GFP-SgA-R	CCGGTGATGCGGCACTCGATCTCGGCCATAATGGCCGTTTAAACGGCGCGCCGGCCTAGGCGGCCATAACTTCGTATAGCATACATTATACGAAGTTATTTAATTAAATAACTTCGTATAGCATACATTATACGAAGTTATCCGGTGATGCGGCACTCGATCTC
SfiIx2-F	CTAGCAAGGCCGCCTAGGCCGGCGCGCCGTTAAACGGCCATTATGGCCGTTT
SfiIx2-R	AAACGGCCATAATGGCCGTTTAACGGCGCGCCGGCCTAGGCGGCCTTG
gUL21/22-F	CACCGGCGTAATTTCATAACCAAA
gUL21/22-R	AAACTTTGGTTATGAAATTACGCC
gUL3.5/4-F	CACCGTTTACGTCCTCGGGGGAGG
gUL3.5/4-R	AAACCCTCCCCCGAGGACGTAAAC
gUL7/8-F	CACCGTATTTCAAATATTAGTGTG
gUL7/8-R	AAACCACACTAATATTTGAAATAC
gUL40/41-F	CACCGTGGTAATCAAGAGTTTACT
gUL40/41-R	AAACAGTAAACTCTTGATTACCAC
gUL55/LORF11-F	CACCGACTATGTAAAGACAGTCGA
gUL55/LORF11-R	AAACTCGACTGTCTTTACATAGTC
gSORF3/US10-F	CACCGTTTGCAATCCTTTATACAT
gSORF3/US10-R	AAACATGTATAAAGGATTGCAAAC
gUL44/44.5-F	CACCGCACAACTTCAAAAATGATG
gUL44/44.5-R	AAACCATCATTTTTGAAGTTGTGC
gUL50/51-F	CACCGCAACCTCTTCATATTAGAT
gUL50/51-R	AAACATCTAATATGAAGAGGTTGTC

**Table 3 vaccines-10-00686-t003:** Primers used for junction PCR and OmpH *-V5 gene inserted verification.

Primer	Sequences
UL55-F	GGCGCGAGAAACTAGTGGT
UL55-R	CGCGCAAAAAGTAAAGACCCA
UL44-F	TTTAGGCGTTTTGCCCGTTC
UL44.5-R	GGCTGGAATTTTAACCGGCG
OmpH-3F	ACGTGCTCTTGAAGTGGGTT
OmpH-5R	GCGAAACCCGCATAAAGACG
Omp-F	CAACAGTTTACAATCAAGAC
OmpH-V5-R	GCGGCCGCTTACGTAGAATCGAGACCGAG

* Genbank accession number: U50907.1.

## Data Availability

All data in this study have been included in the manuscript.

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
