# Peer review of "CRISPR/Cas9 Editing of Duck Enteritis Virus Genome for the Construction of a Recombinant Vaccine Vector Expressing ompH Gene of Pasteurella multocida in Two Novel Insertion Sites"

_vaccines, 2022, doi:10.3390/vaccines10050686_

Round 1
Reviewer 1 Report
In this manuscript, Apinda et al use CRISPR/Cas9 editing to generate a stable live vaccine based on duck enteritis virus (DEV), which is aimed to protect simultaneously against duck plague and fowl cholera. These are worldwide and sometimes fatal diseases of water fowl, and can have serious economic impact. The OmpH gene of Pasteurella multocida is inserted in 9 different intergenic regions of the DEV genome in order to find locations that can stably maintain the OmpH gene. First, they show that locations UL55 and UL44 allow for better expression of an inserted GFP-containing cassette. Next, they generate two live viruses expressing OmpH from these distinct locations, and also expressing GFP. The latter gene is floxed and is then removed by transfecting cells with Cre recombinase. PCR analyses and western blots are used to verify integration and expression of OmpH. Finally plaque size is examined, and multistep growth curves as well as extensive passaging are shown to demonstrate stable expression of the introduced OmpH gene. The paper is clearly written and the data support the conclusions, although some statistical validation would make it more convincing. It is a nice demonstration of the use of CRISPR/Cas9 as an alternative to older techniques such as BAC. CRISPR in combination with Cre-Lox was very similarly applied in previous work with DEV looking for a combinatorial vaccine with and Influenza virus gene. The top insertion site, UL55, was also previously demonstrated as a suitable site. Therefore, although the authors add some stability assessment, the impact of the work as a technical advance is moderate.
Concerns
Without any immune analysis showing that both anti-DEV and anti-OmpH immunity is induced, the technical impact is moderate, for reasons described above.
There is no info anywhere on classification of DEV.
It is not clear how the bar values in Fig 2 are determined, and statistical validation is absent.
Figure 6a and lines 306 and 312: upper and lower panel do not correspond to the text; they appear to be switched .
Figure 6: the data would be more convincing with some statistical validation.
How did the authors select the 9 locations to be tested for insertion?
The authors claim NHEJ-CRISPR editing. How do the authors know that HDR is not involved?
Line 264: Figure 4E should be 4D.
Do the recombinant vaccines still express major DEV antigens?
Although CRISPR/Cas9 is a useful technique for genome modification, there are, in contrast to older modification techniques, indels which cannot be controlled or predicted and can cause variability in gene expression. It would be useful to also discuss disadvantages of the system.
Author Response
Response to reviewers 1
Response to reviewer 1’ comments and suggestions
Comments and Suggestions for Authors
In this manuscript, Apinda et al use CRISPR/Cas9 editing to generate a stable live vaccine based on duck enteritis virus (DEV), which is aimed to protect simultaneously against duck plague and fowl cholera. These are worldwide and sometimes fatal diseases of water fowl, and can have serious economic impact. The OmpH gene of Pasteurella multocida is inserted in 9 different intergenic regions of the DEV genome in order to find locations that can stably maintain the OmpH gene. First, they show that locations UL55 and UL44 allow for better expression of an inserted GFP-containing cassette. Next, they generate two live viruses expressing OmpH from these distinct locations, and also expressing GFP. The latter gene is floxed and is then removed by transfecting cells with Cre recombinase. PCR analyses and western blots are used to verify integration and expression of OmpH. Finally plaque size is examined, and multistep growth curves as well as extensive passaging are shown to demonstrate stable expression of the introduced OmpH gene. The paper is clearly written and the data support the conclusions, although some statistical validation would make it more convincing. It is a nice demonstration of the use of CRISPR/Cas9 as an alternative to older techniques such as BAC. CRISPR in combination with Cre-Lox was very similarly applied in previous work with DEV looking for a combinatorial vaccine with and Influenza virus gene. The top insertion site, UL55, was also previously demonstrated as a suitable site. Therefore, although the authors add some stability assessment, the impact of the work as a technical advance is moderate.
Response: Thank you for the opportunity to address the reviewer’s comments and revise the manuscript accordingly. The authors highly appreciate the reviewers’ insightful and helpful comments on our manuscript. The comments are encouraging. All page numbers refer to the manuscript file. All revisions in the manuscript are marked using the Word “track change” feature based on the comments as described below.
Concerns
Without any immune analysis showing that both anti-DEV and anti-OmpH immunity is induced, the technical impact is moderate, for reasons described above.
1) There is no info anywhere on classification of DEV.
Response: Thank you for this suggestion. We have added detail about DEV classification in introduction paragraph between line 47-54.
2) It is not clear how the bar values in Fig 2 are determined, and statistical validation is absent.
Response: Thank you for this suggestion. The bar values in Fig2 determined from the numbers of GFP-positive green infected cell areas assumed it is the capability and efficacy of sgRNA in insertion. As your suggestion, the reporting p-values of statistical test was added. Please see line 241-242.
3) Figure 6a and lines 306 and 312: upper and lower panel do not correspond to the text; they appear to be switched.
Response: Thank you for pointing this out. We have done to switch this point. Please see line 341 and 347.
4) Figure 6: the data would be more convincing with some statistical validation.
Response: Thank you for pointing this out. The reporting p-values of statistical test was added. Please see line 368.
5) How did the authors select the 9 locations to be tested for insertion?
Response: We sincerely appreciate the reviewer’s comments. Intergenic regions are a subset of noncoding DNA.We used all intergenic regions in DEV genome which a stretch of DNA sequences located between opposite direction (forward and reverse directions) of the genes. Moreover, these 9 locations in the study have never been editing by CRISPR/Cas9. Our results demonstrate which location can be used as a stable site for exogenous sequence insertion by only one gRNA of intergenic region.
The authors claim NHEJ-CRISPR editing. How do the authors know that HDR is not involved?
Response: Thank you for interesting point. Both HDR and NHEJ-CRISPR/Cas9-mediated genome editing technology has been successfully used for genetic engineering of DEV. The previous study has successfully constructed the trivalent DEV vaccine using HDR-CRISPR/cas9 but needed a combination with SCR7 because HDR is less frequent than NHEJ and only occurs during S and G2 phases, whereas NHEJ occurs throughout the cell cycle. Thus, this study selected NHEJ pathway because it also faster and more efficient than HDR.
Line 264 Figure 4E should be 4D.
Response: The authors do apologize for the incorrect word and revised as requested. Please see line 293.
Do the recombinant vaccines still express major DEV antigens?
Response: We sincerely appreciate the reviewer’s comments. Both rDEV-UL55 and UL44 still expressed protein similarly with DEV wild type vaccine strain as figure 6a (Top).
Although CRISPR/Cas9 is a useful technique for genome modification, there are, in contrast to older modification techniques, indels which cannot be controlled or predicted and can cause variability in gene expression. It would be useful to also discuss disadvantages of the system.
Response: Thank you for this suggestion. The limitations of CRISPR/Cas9 were discussed in this revised form of the manuscript. Please see our response at line 487-494.
Reviewer 2 Report
General comments
In this study, the authors identified novel intergenic sites for the insertion of foreign genes in the DEV genome using CRISPR/Cas9-based approaches. This study is generally well-designed and provides invaluable information to develop DEV-based vaccines. However, this reviewer has some comments to improve this paper as follows;
1) Why did the authors focus on only UL regions for searching suitable sites for the insertion of foreign genes. Are there any possibilities that suitable sites are present in other regions?
Also, why did the authors focus on only intergenic regions? How about genes unnecessary for the replication?
2) Are the amounts of antigen sufficient to induce the immune responses for the protection against the disease caused by Pasteurella multocida? To confirm the utility as vaccines, the antibody titer induced by the immunization with these recombinant viruses should be examined by animal experiments. At least, the potentials to induce effective immune responses should be discussed by citing appropriate references.
Other comments
This manuscript includes many grammatical errors. This manuscript needs to be carefully re-edited for English grammar, or needs English proofreading.
Lines 139-140: Please add the detailed information. Did the author sort each cell infected with DEV using FACS or purify the plaques?
Line 141: Please modify to “plaque”
Line 142: “before being infection with” should be modified to “before the infection with”.
LIne 150: Insert a space between % and agar.
Line 157: virus-infected
Line 158: Serum? polyclonal antibody? Please reconsider the description.
Line 159: 6 should be a superscript.
Line 167: 1:5000 dilution
Lines 167-169: Reconsider the sentence.
Line 175: mouse anti-V5 monoclonal antibody
Line 177: DEV-infected cells
Line 188: Table 2? The names of primers were different from those in Figure 4A. It can be a bit confusing. Please use the same names.
Line 190: compared
Line 228: “virus” should be deleted.
Line 229: “will be “ should be modified to “is”.
Line 230: “virus” should be deleted.
Line 234: What is “28”?
Line 235: Delete a hyphen of Sg-A.
Line 248: Please mention the method of FACS in the Materials and Methods section.
Line 262: “2” should be a subscript.
Line 264: Figure 4d?
Line 265: table 2?
Line 268-270: Reconsider the sentence.
Line 275: primers
Line 294: primers
Line 295: previously in Figure 4A
Line 301: What is DEV-GFP-HA?
Lines 301-304: Reconsider the sentence.
Line 306: Lower?
Line 312: Upper?
Lines 317-318: Reconsider the sentence.
Lines 318-319: Reconsider the sentence.
Lines 315-321: The explanations are reversed for upper and lower panels.
Line 322: “of” should be deleted.
Line 326-327, 328: “DEV recombinant virus” should be modified to “recombinant DEVs”. V in DEV means “virus”.
Line 329: were
Line 335: Delete “virus”.
Line 337: Delete “virus”.
Line 338: Were the properties determined by size or morphology? Reconsider the sentence.
Line 346: “7” should be a superscript.
Line 350: recombinant DEV-OmpH-V5.
Lines 361-362. Reconsider the sentence.
Line 363: recombinant DEV
Line 364: assessed
Lines 367-368: Do the authors mean that the insert was lost form the recombinant virus during several passages? But, finally, the authors concluded that the insertion was stably present in the virus genome.
It’s really confusing. Reconsider the sentences and structure of this paragraph.
Line 410: stable expression of foreign genes
Line 411: sites
Lines 410-413: Reconsider the sentence.
Line 415: These results are
Lines 415-417: Reconsider the sentence.
Author Response
Response to reviewers 2
Reviewer 2
General comments
In this study, the authors identified novel intergenic sites for the insertion of foreign genes in the DEV genome using CRISPR/Cas9-based approaches. This study is generally well-designed and provides invaluable information to develop DEV-based vaccines. However, this reviewer has some comments to improve this paper as follows;
Response to reviewer 2’ general comments
Thank you for giving me the opportunity to submit a revised draft of my manuscript. I appreciate the time and effort that you and the reviewers have dedicated to providing your valuable feedback on my manuscript. All page numbers refer to the manuscript file. I have been able to incorporate changes to reflect the suggestions provided by the reviewers. All revisions in the manuscript are marked using the Word “track change” feature based on the comments as described below.
Reviewer 2’ comments and Suggestions for Authors
1) Why did the authors focus on only UL regions for searching suitable sites for the insertion of foreign genes. Are there any possibilities that suitable sites are present in other regions?
Response: Thank you for pointing this out. I used all intergenic regions in DEV genome which a stretch of DNA sequences located between opposite direction (forward and reverse direction) of the genes. These 9 locations (including US region) in the study have never been editing by CRISPR/Cas9. Our results demonstrated which location can be used as a stable site for exogenous sequence insertion by only one gRNA of intergenic region.
2) Also, why did the authors focus on only intergenic regions? How about genes unnecessary for the replication?
Response: The authors would like to thank the reviewer for interesting point. Previous study shown that some intergenic region in DEV genome was proved as a suitable site for insertion foreign gene by several technique including CRISPR/Cas9. This study has been confirmed and found out for other locations as a novel stable site. About other genes unnecessary for the replication including miRNA gene which are very interesting for future studies.
3) Are the amounts of antigen sufficient to induce the immune responses for the protection against the disease caused by Pasteurella multocida? To confirm the utility as vaccines, the antibody titer induced by the immunization with these recombinant viruses should be examined by animal experiments. At least, the potentials to induce effective immune responses should be discussed by citing appropriate references.
Response: The authors are grateful for this comment as it points to an important rationale of this study, which confirmed the utility of OmpH expression as vaccines against P. multocida. However, the authors would further consider performing DEV and P. multocida challenge post immunization with our recombinant vaccine in animal experiment to obtain complete comprehension. However, in the case of this study, the main objective was identified novel intergenic sites for the insertion of foreign genes in the DEV genome using CRISPR/Cas9. Thus, the cellular and humoral response will be clearly examined in our future study. As suggested by reviewer, the potential of ompH gene to induce effective immune responses in duck from previous study was discussed and cited at line 426-430.
Other comments
1) This manuscript includes many grammatical errors. This manuscript needs to be carefully re-edited for English grammar, or needs English proofreading.
Response: We regret there were problems with the grammatical errors. We have polished this manuscript to improve the grammar and readability.
2) Lines 139-140: Please add the detailed information. Did the author sort each cell infected with DEV using FACS or purify the plaques?
Response: We sincerely appreciate your comments. Yes, we use FACS to get only single green cell and future purify plaque by the recombinant post-sorting. As your suggestion, we have added the suggested content to the manuscript on material & method part between line 154-163.
Line 141: Please modify to “plaque”
Response: The authors do apologize for the incorrect word. The corrected word was modified in the revised manuscript at line 149.
Line 142: “before being infection with” should be modified to “before the infection with”.
Response: Revised as requested. Please see at line 150.
LIne 150: Insert a space between % and agar.
Response: Revised as requested. Please see at line 169.
Line 157: virus-infected
Response: Revised as requested. Please see at line 180.
Line 158: Serum? polyclonal antibody? Please reconsider the description.
Response: Revised as requested. Please see at line 181.
Line 159: 6 should be a superscript.
Response: Revised as requested. Please see at line 183.
Line 167: 1:5000 dilution
Response: Revised as requested. Please see at line 190.
Lines 167-169: Reconsider the sentence.
Response: I have revised the sentence. Please see at line 191-194.
Line 175: mouse anti-V5 monoclonal antibody
Response: Revised as requested. Please see at line 200.
Line 177: DEV-infected cells
Response: Revised as requested. Please see at line 203.
Line 188: Table 2? The names of primers were different from those in Figure 4A. It can be a bit confusing. Please use the same names.
Response: The authors do apologize for this confusing. The rename primers and details were added into table 3. Please see at line 176-177.
Line 190: compared
Response: Revised as requested. Please see at line 215.
Line 228: “virus” should be deleted.
Response: Correction has been made. Please see at line 257.
Line 229: “will be “ should be modified to “is”.
Response: Revised as requested. Please see at line 258.
Line 230: “virus” should be deleted.
Response: Correction has been made. Please see at line 259.
Line 234: What is “28”?
Response: The authors do apologize for this mistake. 28 is number of reference list. It was added with [ ]. Please see at line 263.
Line 235: Delete a hyphen of Sg-A.
Response: Correction has been made. Please see at line 264.
Line 248: Please mention the method of FACS in the Materials and Methods section.
Response: As your suggestion, we have added the suggested content to the manuscript on material & method part between line 154-163.
Line 262: “2” should be a subscript.
Response: Revised as requested. Please see at line 291.
Line 264: Figure 4d?
Response: The authors do apologize for this confusing. The correction has been changed to Figure 4d at line 293.
Line 265: table 2?
Response: The authors do apologize for this confusing. The purification primers were separated and added into table 3. Please see at line 176-177.
Line 268-270: Reconsider the sentence.
Response: We have revised the sentence. Please see at line 297-299.
Line 275: primers
Response: Revised as requested. Please see at line 298.
Line 294: primers
Response: Revised as requested. Please see at line 306
Line 295: previously in Figure 4A
Response: Revised as requested. Please see at line 326.
Line 301: What is DEV-GFP-HA?
Response: The authors do apologize for this confusing. Correction has been made. Please see at line 332.
Lines 301-304: Reconsider the sentence.
Response: We have revised the sentence. Please see at line 332-336.
Line 306: Lower?
Response: Correction has been made. Please see at line 341.
Line 312: Upper?
Response: Correction has been made. Please see at line 347.
Lines 317-318: Reconsider the sentence.
Response: We have revised the sentence. Please see at line 352-355.
Lines 318-319: Reconsider the sentence.
Response: We have revised the sentence. Please see at line 352-355.
Lines 315-321: The explanations are reversed for upper and lower panels.
Response: Correction has been made. Please see at line 351 and 359.
Line 322: “of” should be deleted.
Response: Correction has been made. Please see at line 360.
Line 326-327, 328: “DEV recombinant virus” should be modified to “recombinant DEVs”. V in DEV means “virus”.
Response: Revised as requested. Please see at line 365-366.
Line 329: were
Response: Correction has been made. Please see at line 367.
Line 335: Delete “virus”.
Response: Revised as requested. Please see at line 373.
Line 337: Delete “virus”.
Response: Revised as requested. Please see at line 375.
Line 338: Were the properties determined by size or morphology? Reconsider the sentence.
Response: The authors do apologize for this confusing. Correction has been made at line 376.
Line 346: “7” should be a superscript.
Response: Revised as requested. Please see at line 384.
Line 350: recombinant DEV-OmpH-V5.
Response: Revised as requested. Please see at line 390.
Lines 361-362. Reconsider the sentence.
Response: We have revised the sentence. Please see at line 400-402.
Line 363: recombinant DEV
Response: Revised as requested. Please see at line 404-405.
Line 364: assessed
Response: Correction has been made. Please see at line 405.
Lines 367-368: Do the authors mean that the insert was lost form the recombinant virus during several passages? But, finally, the authors concluded that the insertion was stably present in the virus genome.
It’s really confusing. Reconsider the sentences and structure of this paragraph.
Response: The authors do apologize for this confusing. Early, I would like to mean if the insert was loss from any of the recombinant viruses, the staining would only show positive labelling with the anti-DEV antibody. As your suggestion, I have revised clearer this paragraph, please see between line 408-416.
Line 410: stable expression of foreign genes
Response: Revised as requested. Please see at line 465.
Line 411: sites
Response: Correction has been made. Please see at line 467.
Lines 410-413: Reconsider the sentence.
Response: We have revised the sentence. Please see at line 466-469.
Line 415: These results are
Response: Correction has been made. Please see at line 474.
Lines 415-417: Reconsider the sentence.
Response: We have revised the sentence. Please see at line 474-477.
Reviewer 3 Report
The manuscript submitted by Apinda et al. entitled "CRISPR/Cas9 editing of duck enteritis virus genome for the construction of a recombinant vaccine vector expressing ompH gene of Pasteurella multocida in two novel insertion sites" aims to report for the first time the use of the CRISPR/Cas9 system to insert a highly immunogenic gene from bacteria into the Duck enteritis virus genome in a rapidly and efficiently way. The results reported are very promising since DEV-OmpH-V5-UL55-LORF11 and DEV-OmpH-V5- UL44-44.5 mutants, with the insertion of ompH-V5 gene at the UL55-LORF11 and UL44-44.5 loci respectively, showed similar growth kinetics and plaque size, compared to the wildtype virus, confirming that the insertion of foreign gene into these did not have any detrimental effects on DEV. Additionally, the M&M are very well presented and results discussed in a correct and pragmatic way. Figures and tables are also nice presented. Thus, in the opinion of this reviewer, the manuscript is suitable for publication in the present form.Author Response
Response to reviewers 3
Comments and Suggestions for Authors
The manuscript submitted by Apinda et al. entitled "CRISPR/Cas9 editing of duck enteritis virus genome for the construction of a recombinant vaccine vector expressing ompH gene of Pasteurella multocida in two novel insertion sites" aims to report for the first time the use of the CRISPR/Cas9 system to insert a highly immunogenic gene from bacteria into the Duck enteritis virus genome in a rapidly and efficiently way. The results reported are very promising since DEV-OmpH-V5-UL55-LORF11 and DEV-OmpH-V5- UL44-44.5 mutants, with the insertion of ompH-V5 gene at the UL55-LORF11 and UL44-44.5 loci respectively, showed similar growth kinetics and plaque size, compared to the wildtype virus, confirming that the insertion of foreign gene into these did not have any detrimental effects on DEV. Additionally, the M&M are very well presented and results discussed in a correct and pragmatic way. Figures and tables are also nice presented. Thus, in the opinion of this reviewer, the manuscript is suitable for publication in the present form.
Response to reviewer 1’ comments and suggestions
Authors are grateful to the reviewer for giving us the opportunity to submit a revised draft of our manuscript titled “CRISPR/Cas9 editing of duck enteritis virus genome for the construction of a recombinant vaccine vector expressing ompH gene of Pasteurella multocida in two novel insertion sites" to this journal.WeI appreciate the time and effort that you have dedicated to providing your valuable feedback on my manuscript. All your comments are positive and encouraging to us.